# First Molecular Detection and Epidemiological Analysis of Equine Influenza Virus in Two Regions of Colombia, 2020–2023

**DOI:** 10.3390/v16060839

**Published:** 2024-05-24

**Authors:** Juliana Gonzalez-Obando, Angélica Zuluaga-Cabrera, Isabel Moreno, Jaime Úsuga, Karl Ciuderis, Jorge E. Forero, Andrés Diaz, Carlos Rojas-Arbeláez, Juan P. Hernández-Ortiz, Julian Ruiz-Saenz

**Affiliations:** 1Grupo de Investigación en Ciencias Animales—GRICA, Facultad de Medicina Veterinaria y Zootecnia, Universidad Cooperativa de Colombia, Bucaramanga 680002, Colombia; juliana.gonzalezo@udea.edu.co; 2Grupo de Epidemiología, Universidad de Antioquia, Medellín 050010, Colombia; carlos.rojas@udea.edu.co; 3Grupo de Investigación GISCA, Facultad de Medicina Veterinaria y Zootecnia, Fundación Universitaria Vision de las Américas, Medellín 050031, Colombia; angelica.zuluagac@uam.edu.co; 4GHI One Health Colombia, Universidad Nacional de Colombia, Medellín 050036, Colombia; imorenol@unal.edu.co (I.M.); jausugar@unal.edu.co (J.Ú.); coord_cwohc@unal.edu.co (K.C.); jphernandezo@unal.edu.co (J.P.H.-O.); 5Grupo de Investigación en Microbiología Ambiental, Escuela de Microbiología, Universidad de Antioquia, Medellín 050010, Colombia; jorge.forero@udea.edu.co; 6Pig Improvement Company Hendersonville, Hendersonville, TN 37075, USA; andres.diaz@genusplc.com

**Keywords:** epidemiology, molecular diagnostic, Colombia, risk factors, horses, equine influenza

## Abstract

Equine influenza is a viral disease caused by the equine influenza virus (EIV), and according to the WOAH, it is mandatory to report these infections. In Latin America and Colombia, EIV risk factors have not been analyzed. The objective of this research is to perform an epidemiological and molecular analysis of the EIV in horses with respiratory symptoms from 2020 to 2023 in Colombia. Molecular EIV detection was performed using RT–qPCR and nanopore sequencing. A risk analysis was also performed via the GEE method. A total of 188 equines with EIV respiratory symptoms were recruited. The positivity rate was 33.5%. The descriptive analysis showed that only 12.8% of the horses were vaccinated, and measures such as the quarantine and isolation of symptomatic animals accounted for 91.5% and 88.8%, respectively. The variables associated with the EIV were the non-isolation of positive individuals (OR = 8.16, 95% CI (1.52–43.67), *p* = 0.014) and sharing space with poultry (OR = 2.16, 95% CI (1.09–4.26), *p* = 0.027). In conclusion, this is the first EIV investigation in symptomatic horses in Colombia, highlighting the presence of the virus in the country and the need to improve preventive and control measures.

## 1. Introduction

Equine influenza, which is caused by the equine influenza virus (EIV), is a highly contagious disease distributed worldwide [1,2,3], and cases must be reported to the World Organization for Animal Health (WOAH) [2,3]. The EIV belongs to *Orthomyxoviridae*; the genus name is *Alphainfluenzavirus*, and the species is *Alphainfluenzavirus influenzae*. [4]; it has an enveloped capsid and contains eight negative sense genomic segments of single-stranded RNA. The surface proteins hemagglutinin (HA) and neuraminidase (NA) determine the EIV subtype [5,6,7]. Currently, only EIV H3N8 causes significant economic losses in the equine industry [2].

Although a preference for α-2,3 sialic acid receptors is shown by equine influenza viruses [8,9], different evolutionary analyses have proposed an avian origin for the EIV [10,11,12]. Furthermore, the EIV has been documented to jump species barriers and infect various animal species, such as cats, pigs, and dogs [13,14]. This phenomenon is facilitated by shared glycan receptors possessing sialic acid residues in α-2,3 bonds, enabling cross-species infection. Cases of both spontaneous infections and experimental infections in animal models have been reported, as evidenced by the 2004 case of influenza transmission from equines to canines, which spread extensively throughout the United States [15]. Similar occurrences in felines have also been documented through natural infections and experimental models [16,17].

The clinical signs of equine influenza include fever, enlarged lymph nodes, nasal discharge, and cough [2,18]. Vaccination may reduce clinical symptoms [19,20]. The transmission of the EIV between horses is associated with low vaccination rates, inadequate isolation practices, attendance at competitions, and delayed diagnosis [18,20,21,22]. However, the epidemiology and risk factors for the EIV in the country and other Latin American countries are poorly understood.

In Colombia, the first equine influenza case was reported in 1982, followed by outbreaks in the 1990s and 2000s [18,23]. Additionally, since 1999, EIV vaccination has been mandatory for mobilizing horses for equestrian exhibitions or competitions [24]. Although EIV outbreaks were reported in South America in 2018 and 2022 [18,23], the current circulation of the EIV in Colombia is unknown. Therefore, the objective of this study is to assess the presence of the EIV in Colombia using molecular diagnostics and to explore the risk factors and symptoms associated with the disease in the country.

## 2. Materials and Methods

A descriptive cross-sectional study was carried out in equines from two regions in Colombia (the Departments of Antioquia and Cundinamarca) from November 2020 to April 2023. Antioquia and Cundinamarca are the regions (Appendix A) with the largest number of horses registered in the country and where most equestrian activities and horse-related events occur [25,26].

A convenience sampling approach was used in this study. Domestic equines that met the inclusion criteria were selected and sampled. The inclusion criteria were Equidae species (horses, mules, and/or donkeys) of all ages and of any sex who presented with symptoms of influenza-like illness (bilateral nasal discharge, cough, and positive cough reflex) and who also lived at the selected sites (Appendix A).

### 2.1. Epidemiological Variables: Survey

A written questionnaire of 37 elements (survey form, Appendix A) was used to record epidemiological variables and other characteristics, including individual information on the animal, management and animal health practices, and location, including animal characteristics related to age, sex, type of activity, and breed. The clinical characteristics and treatment included fever (higher than 38.5 °C); decreased performance; increased size of the retropharyngeal, cervical, and parotid lymph nodes; reduced food consumption; loss of weight; respiratory distress; runny nose; cough; eye discharge; and treatment. The herd characteristics included the number of stables, number of occupied stables, number of equines (horses, mules, and donkeys), and predominant breed in the herd. The health management variables and possible risk factors include coexistence with other animal species, such as canines, felines, pigs, and others, due to the possible risk of infection between species by EIV subtype H3N8, vaccination, type of vaccine, time of the first vaccination, vaccination of pregnant mares, type of deworming, frequency of deworming, product with which the disinfection of the herds is carried out, entry of new horses in the last year, location from which they enter, clinical records, types of medical records (written or virtual), participation of the herds in massive events, and number of horses sampled. This survey was administered to the caregivers and veterinary doctors in charge.

### 2.2. Sample Collection

Nasopharyngeal swabs (NS) were taken from 188 equines with respiratory symptoms, mainly cough and nasal discharge, in 40 different equine herds (*n* = 32/40 in Antioquia and *n* = 8/40 in Cundinamarca) using a Medline^®^ Swab with a 16′ Rayon Tip (Medline Industries, LP. Northfield, IL, USA). After collection, the swab tips were stored in viral transport media and frozen at −80 °C until RNA extraction.

### 2.3. Molecular Diagnosis and H3N8 Confirmation

RNA extraction was performed with a QIAmp RNA Viral Kit and an RNA Easy Blood and Tissue Kit (Qiagen Inc., Valencia, CA, USA) following the manufacturer’s instructions. The quality and quantity of RNA were determined using a Nanodrop^®^ ONE (Thermo Fisher Scientific, Rockford, IL, USA). Then, the obtained RNA was stored at −80 °C until RT–qPCR analysis.

Reverse transcription was performed using the RevertAid First Strand cDNA Synthesis Kit (Thermo Scientific^®^ Waltham, MA, USA) following the manufacturer’s instructions, and cDNA was stored at −20 °C. Then, qPCR was performed as previously described by Heine et al. (2007) [27], but a TaqMan probe, directed against the matrix proteins of the influenza virus type A, was used [28]. The TaqMan™ Gene Expression Master Mix Kit (Applied Biosystems™ Life Technologies, Darmstadt, Germany) was used in the QuantStudio™ Real-Time PCR System (Applied Biosystems^®^) following the manufacturer’s instructions. CT values < 38 were considered positive. H3N8-positive cDNA, which was generated by Stephanie Reedy from the University of Kentucky, was used as a positive control, and ddH2O was used as a negative control.

For the rt-qPCR-positive samples, multisegment RT–PCR was performed by using a SuperScript™ III One-Step RT–PCR System with Platinum™ Taq DNA Polymerase (Invitrogen, Life Technologies, Carlsbad, CA, USA) and using influenza-specific universal primers at the end of all 8 genomic segments, as previously reported [29]. Only three samples showed a clear electrophoresis band pattern. Positive samples were analyzed using sequencing performed on the Minion platform (Oxford Nanopore sequencing) [30,31]. The amplicons were purified using Beckman Coulter™ Agencourt AMPure XP (Beckman Coulter, Brea, CA, USA) with an individual barcode for each sample by using the SQK-LSk109 and EXP-NBD196 kits (Oxford Nanopore Technologies, Oxford, UK). The library was loaded on an R9.4 Oxford MinION flowcell and sequenced using a MinION Mk1B device. Base-calling and demultiplexing were performed using Guppy (v6.0.1).

The reference genome used for homology-based assembly was A/equine/Ohio/OH21-6023/2021 (H3N8). HA sequences were registered in GenBank under accession numbers PP737049–PP737051. For the phylogenetic analysis, 93 representative HA sequences were obtained from each EIV sublineage and the sequences in our study. A maximum-likelihood phylogeny was constructed by using the stringent GTR + G algorithm, which was identified by using the best model tool available in MEGA 7, by bootstrapping with 1000 replications.

### 2.4. Data Analysis

The analysis was carried out with the statistical program SPSS ver. 21 (IBM Corporation), which is licensed. A descriptive analysis of the collected data was carried out, including the mean, median, standard deviation, and range. For qualitative variables, absolute and relative frequencies were used. We used tabular methods to estimate the crude association between exposure variables and the detection of the EIV using RT–qPCR (positive or negative) in horses with respiratory symptoms and considered the association to be significant when the *p* value for the chi-square test or Fisher’s exact test was lower than 0.05 with a 95% confidence level. Subsequently, only epidemiological variables that had a value of 0.25 or less were ultimately chosen for inclusion in the GEE (Generalized Estimated Equation) model, with the equine herd as a cluster (random effects) to estimate the adjusted association between the EIV and RT–qPCR detection of other exposures. For the analysis of the model, all variables were dichotomized into zeros and ones for model fitting. Additionally, all models demonstrated an improved model fit and parsimony using QIC [32,33,34,35]. In addition, a bivariate analysis was performed in order to detect variables that had an association with the confirmed molecular diagnosis, using the chi-square test or Fisher test, and a *p* value below 0.05 with a 95% confidence level was considered an association.

### 2.5. Ethical Statement

We followed all the international and Colombian ethical standards for biomedical research with animal subjects established by bioethical and animal research. The study was reviewed and approved by the Ethics Committee for Animal Experimentation of the Universidad Cooperativa de Colombia in Bucaramanga (Minute number 044-2018). Written informed consent was obtained from the animal owners.

## 3. Results

A total of 188 equines were tested from 40 equine herds between 2020 and 2023, and these herds were analyzed only once. Eight herds were from Cundinamarca, and thirty-two were from Antioquia, with 36% and 64% of the animals tested, respectively. Sixty-three (33.5%) of the strains tested were positive according to RT–qPCR, and all were detected in 2022 (Table 1).

### 3.1. Molecular Diagnosis and Phylogenetic Analysis

The overall percentage of positive samples obtained using RT–qPCR was 33.5% (63/188). The positivity rate in Antioquia was 35.5% (43/121), while that in Cundinamarca was 29.8% (20/67). From the three sequenced samples, all of which belonged to the Antioquia Department, we confirmed the presence of the H3N8 subtype in Florida clade 1. Our sequences formed a monophyletic clade with sequences from the United States with a bootstrap value of 98. These results are also related to the sequences reported from the southern cone outbreak in 2018 (Figure 1).

### 3.2. Zoographic Characteristics

Ninety-nine percent of the samples were from horses (186/188), and 1% were from mules (2/188). A total of 82.4% (155/188) were Colombian Creole horses, and 71.8% (135/188) were female (Appendix A). The median age was three years, and the IQR was twenty-three. When the rt-qPCR results were distributed by age, it was possible to find that the age group with the greatest number of positive animals was <2 years old (8/27). According to sex, females were the group with the most positive cases (36.3%, 49/135), and activity competition was the group with the greatest number of positive animals (53.3%—16/30) However, for these differences by groups (age/sex/activity), no statistical difference was found (Table 2).

### 3.3. Bivariate Analysis of Clinical Variables and Treatment

Clinical severity variables in animals positive for the EIV were bilateral nasal discharge 95.2% (60/63), dry cough 93.7% (*n* = 59/63), enlarged retropharyngeal nodes 52.4% (33/63), and fever 41.3% (26/63), These last three had a statistically significant difference. Infrequent symptoms in positive animals included respiratory distress 7.9% (5/63), eye discharge 3.2 (2/63), and productive cough 7.9% 5/63 (Table 3). A statistical association was found in the bivariate analysis of the following variables: dry cough, fever, and enlarged retropharyngeal lymph nodes. In the veterinary treatment, the use of mucolytics were the most commonly used therapeutic strategy in positive animals 41.3% (26/63); however, 42.9% of the equines did not receive any treatment (27/63).

### 3.4. Characteristics of the Horse Herds

The average number of horses per herd was 56.18 (rank 2–240). The average number of stalls per herd was 53.77 (2–200), with 52.57 stalls being occupied (2–200). Stables housed all of the horses (188/188).

### 3.5. Shared Space with Other Species

Considering the well-known interspecies transmission of the EIV to various animal species, we examined the presence of different animals on the premises (Figure 2). Our findings revealed that canines constituted the predominant animal species sharing space with the horses under study, accounting for 96.8% (182 out of 188) of the horses. A similar trend was observed among the EIV-positive animals. Additionally, cats (60.3%) and poultry (41.3%) were identified as other significant species in close proximity to the EIV-positive equines. However, none of these animals (dogs, cats, poultry) exhibited clinical signs of influenza during the study period.

The category “sharing space with poultry” exhibited a statistically significant association (odds ratio = 2.16, 95% confidence interval 1.09–4.26, *p* = 0.027), possibly related to poor biosecurity practices in herds.

### 3.6. Vaccination and Sanitary Measures

Thirteen percent (24/188) of the horses sampled in the study were vaccinated against the EIV in the previous year, and 15.9% (*n* = 10) of the EIV-infected animals had been vaccinated. When we analyzed the type of activity and vaccination status of the studied animals, we found that those animals that required movement, such as shows, exhibitions, competitions, and saddle types (136/188), did not have an updated vaccination schedule for equine influenza at the time of sampling.

### 3.7. Other Sanitary Measures Evaluated

According to our survey, important preventive interventions were not applied in most of the herds. No separation of healthy and sick equines was performed in most of the patients (88.8%, 167/188), 33.1% of whom were positive (62/167). Furthermore, 89.4% (168/188) of the horses were housed on properties where quarantine was not required for new horse arrival.

### 3.8. Herd Variables and Disinfection and Health Management

Overall, sodium hypochlorite (bleach) was the most commonly used disinfectant (27.1%, 51/188). Glutaraldehyde and hydrated lime were the disinfectants with the lowest percentages, at 9.0% (17/188) and 3.2% (6/188), respectively (Table 4).

### 3.9. Veterinary Care and Health Management

We found that 54.8% (*n* = 103/188) of the animals sampled were housed on properties where they only received veterinary assistance if a problem arose. The remaining herds received veterinary care full time. In terms of positivity, 65.1% (*n* = 41/63) of the positive animals lived in herds without a full-time veterinarian.

### 3.10. Sanitary Management Analysis at the Herd Level

Quarantine: in total, 87.5% (35/40) of the horse herds did not require quarantine for new arrivals or had a separate area for new horses that came from another property or after an event. Sick and healthy animals were separated in 85.5% (34/40) of the herds evaluated. Among the properties in which there were positive individuals, 90.5 (19/21) did not quarantine the new individuals who entered (Figure 3).

Variables related to vaccination: fifty percent (20/40) of the herds vaccinated their horses against equine influenza. However, 77.5% (31/40) of the herds did not vaccinate pregnant females against the EIV, and 37.5% (15/40) started the vaccination program after 6 months of age. Twenty-five percent of the herds (10/40) only vaccinated their animals before a competition. When analyzing the positive farms, 85.7% (18/21) of these farms did not vaccinate against the EIV.

Participation in equestrian events: 65% of the herds (26/40) participated in equestrian events (jumping competitions, exhibitions, horseback riding) and international events, e.g., in the United States and Costa Rica. A total of 76% (16/21) of the participants with positive properties reported that equines participated in equestrian events.

In total, 67.5% of the property-generated horses were horses from other geographical areas (27/40), 72.5% (19/27) were from nearby towns, and 27.5% (8/27) were from other countries, including horses from the United States, Costa Rica, and Venezuela. At the herd level, 71.4% (15/21) of the positive herds reported the presence of new horses entering the property.

### 3.11. Multivariate Analysis

Two variables were associated with EIV status (positive or negative) in horses with respiratory disease in this study (Table 3). The odds of testing positive for the EIV were greater in horses housed in herds that did not separate sick animals from healthy individuals and those housed in herds that allowed contact with poultry. The odds of testing positive for the EIV were eight times greater in horses housed in herds where sick animals were not separated from healthy individuals after adjusting for the effects of fever, enlarged lymph nodes, and shared space with poultry (Table 5 and Appendix A).

## 4. Discussion

In the Latin American context, this is the first study looking for variables associated with EIV infection in sick horses. In addition, in Colombia, this is the first study to confirm the H3N8 subtype in horses. The percentage of positive samples was 33.5% (63/188) in symptomatic animals, which might be high for a country in which vaccination has been mandatory for more than 25 years. Our study is the first to confirm the circulation of the H3N8 subtype in the Colombian equine population through next-generation sequencing.

One of the variables associated with EIV detection was a lack of ability to separate sick and healthy horses, with OR = 8.16, in horses belonging to herds where horses with respiratory symptoms were not separated from healthy horses. This finding is consistent with the results of Amjad Khan et al. [36], who reported that a greater number of horses, which made it impossible to separate sick from healthy horses, were infected with equine influenza virus [36], suggesting the need to separate symptomatic equines from susceptible and asymptomatic equines. In addition, isolation should be accompanied by twice-daily body temperature monitoring [37]. Clinical variables with greater severity, such as fever, dry cough, and increased lymph node size, were strongly associated with the presence of the virus. This is strongly expected because fever is an immune system response to the presence of the virus [2], increased lymph node size has been strongly associated with the cellular immune response in EIV-infected horses [20], and dry cough has been reported as one of the most frequent symptoms that persists over time [5]. These three clinical variables could have great predictive value for suspected cases of the EIV.

The high positivity rate in our sample could be explained by the fact that the studied population consisted of symptomatic individuals [20] and that the sampling period coincided with the 2021–2022 outbreak, which had a global distribution, largely in America, Europe, and Asia [7]. Furthermore, in 2022, there was an increase in rainfall in Colombia, mainly between April and May [38], the months in which the outbreak was reported, which may explain the high degree of virus transmission because sick and healthy horses have closer contact, mostly with young susceptible individuals [39,40]. Furthermore, humid and rainy environments are widely recognized to promote outbreaks of influenza A in regions with low latitudes, such as tropical and subtropical zones such as Colombia. This is attributed to the effective transmission of the virus through large droplets and/or aerosols, coupled with the potential for virus survival aided by salts and proteins present in respiratory droplets during humid and rainy conditions. This assertion is supported by extensive research into the survival capabilities of influenza viruses in aerosols, revealing that the maximum survival duration in droplets varies between 1 and 24 h, depending on the relative humidity and the specific influenza strain [41].

Approximately 66.5% of horses exhibiting respiratory symptoms and testing negative for the EIV might have been affected by alternate viral agents such as adenovirus or herpesvirus, both of which are considered potential alternative diagnoses for the EIV [40,42,43,44]. Otherwise, it is possible that the infectious and shedding period of the EIV had elapsed.

Equestrian shows and exhibitions were reopened globally and locally after the SARS-CoV-2 pandemic. Fifteen out of sixty-three (23.8%) horses positive for the EIV were in herds that participated in equestrian shows in the United States. The positive samples found in Colombia could be related to a multifactorial outbreak of respiratory disease occurring in the Americas [45]; however, further sequencing and phylo-evolutionary analysis are needed to better understand the origins of the EIV in Colombia. Previous outbreaks were also reported in the US between 2020 and 2021 [46]. It can be speculated that the EIV was introduced to Colombia by individuals who took part in those international shows, but further sequencing data are needed to support or rule out this hypothesis. The association between EIV infection and horse movement between countries has been previously documented [21,22,23].

Vaccination reduces the likelihood of infection for horses that require mobilization in the country. However, in this study, low vaccination coverage (12.8%) was found. Low vaccination rates or a lack of immunity can be associated with the rapid spread of the EIV, similar to the situation in Chile in 2018, which was related to low herd immunity [47], and Canada [48] and the United States [39]. Low vaccination rates not only favor virus transmission but also may allow for the introduction of new strains [49].

Nevertheless, we found that 15.9% of vaccinated individuals were positive for the EIV. The vaccine used in those animals (Equilis^®^ Prequenza Te—MSD Animal Health, Boxmeer, The Netherlands) was inactivated, and it included viruses from both clades 1 and 2 of the Florida sublineage. Positive cases in vaccinated equines could be associated with virus evolution (antigenic drift) or noncompliance with the entire vaccination protocol, as has been reported in Latin America, Italy, and Croatia [50,51]. As reported by Oladunni et al. in 2021 [20], the constant evolution of viruses, vaccine breakdown, and vaccination-induced short-lived immunity have become constant challenges for achieving full protection against EIV-induced disease, highlighting the need for the constant study of EIV evolution and the host immune response [1,52].

We also found a statistically significant association between being positive for the EIV and living with poultry (OR = 2.16). This could be because EIV-positive horses live in herds with low levels of biosecurity, favoring the coexistence of various animal species in herds [2,53]. It has been suggested that practices involving the cohabitation of mixed species sharing specific receptors, coupled with the absence of segregation between infected and healthy individuals, could facilitate the local dissemination of the equine influenza virus and its transmission among susceptible populations [54,55].

## 5. Conclusions

This study represents the first investigation of equine influenza in Colombia, a country renowned for its equine industry, confirming the presence of the EIV in symptomatic individuals across two regions of the nation. This is the first instance of confirmation through next-generation sequencing of the EIV H3N8 subtype. Furthermore, our findings highlight the absence of adequate sanitary measures that facilitate the presence and dissemination of the virus. These measures include the failure to segregate infected and healthy individuals, low rates of vaccination, and the coexistence of various susceptible species within herds, such as poultry and dogs.

## Figures and Tables

**Figure 1 viruses-16-00839-f001:**
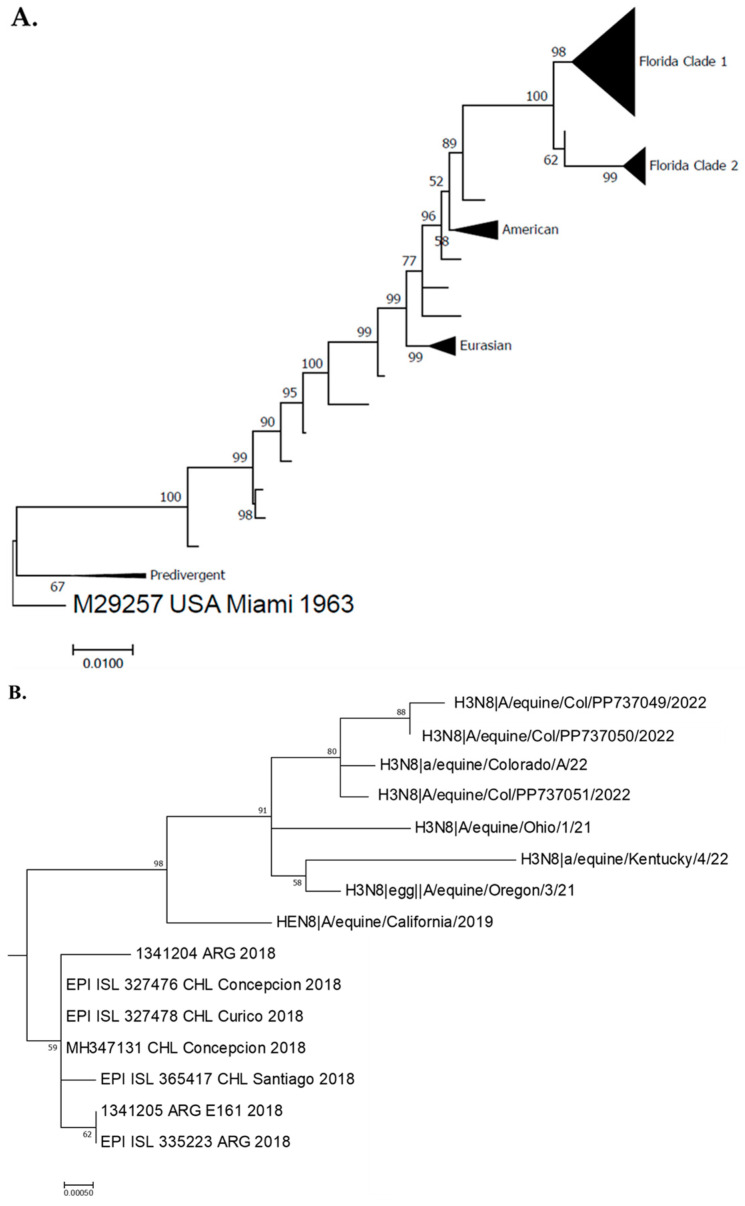
EIV Phylogenetic analysis. (**A**): Maximum likelihood for hemagglutinin (HA) gene nucleotide sequences from 93 equine influenza viruses (EIVs) encoded by the H3N8 subtype of the EIV and Colombian sequences. Bootstrap values obtained after 1000 replicates showing the main clades. Full tree. (**B**): Subtree showing Florida clade 1 clade highlighting Colombian (this study), Chilean, Argentinean, and United states sequences, with bootstrap values obtained after 1000 replicates.

**Figure 2 viruses-16-00839-f002:**
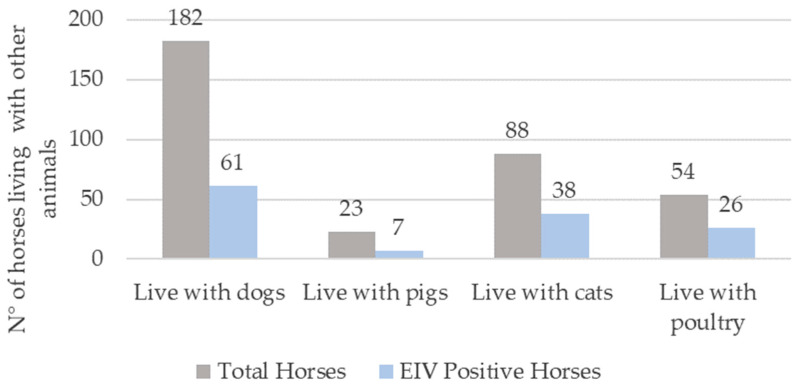
Coexistence of horses with other animal species susceptible to influenza A viruses. The gray bars indicate the total number of horses that cohabit with dogs, pigs, cats, and poultry. The blue bar indicates EIV-positive horses living with different animals.

**Figure 3 viruses-16-00839-f003:**
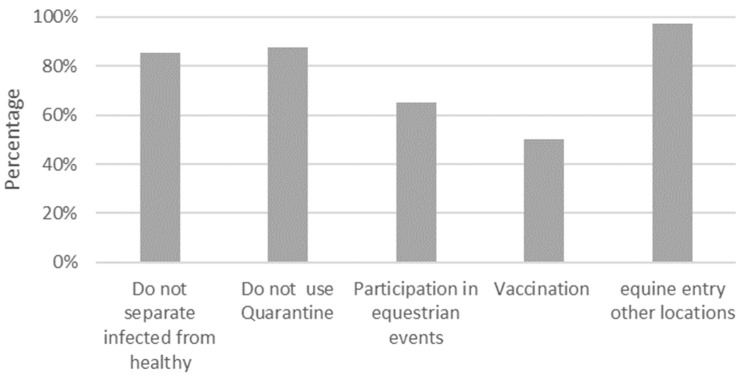
Sanitary measures and health management at herd level. Percentages of the total herds.

**Table 1 viruses-16-00839-t001:** Summary of the equine samples and herds included in the study and percentages according to department and year.

Year	Antioquia	Number of Herds of Antioquia	Positive Herds	Cundinamarca	Number of Herds of Cundinamarca	Positive Herds
2020	0/21 (0%)	8	0	0	0	0
2021	0/38 (0%)	9	0	0	0	0
2022	43/59 (72%)	14	14	20/67 (29%)	8	7
2023	0/3 (0%)	1	0	0	0	0
	121	32	14	67	8	7

**Table 2 viruses-16-00839-t002:** Summary of positivity according to age, sex, and vaccination state categories.

Age (Years)	*n*	Positive	%	*p*-Value
<2	67	29	43.28	0.68
2–5	56	15	26.79	
>5–10	38	11	28.95	
>10	27	8	29.6	
Total	188	63		
**Sex**				**0.20**
Male	53	14	26.4	
Female	135	49	36.3	
**Type of activity**				
Competition	30	16	53.3	0.12
No competition	158	47	29.7	

**Table 3 viruses-16-00839-t003:** Summary of the clinical variables and statistical associations according to bivariate analysis.

Variables	*n*	Positive	%	Negative	%	*p*-Value
**Bilateral Nasal Discharge**						
Yes	181	60	95.2	121	96. 8	0.59
No	7	3	4.8	4	3.2	
**Dry Cough**						
Yes	155	59	93.7	96	76.8	**0.04 ***
No	33	4	6.3	29	23.2	
**Productive cough**						
Yes	13	5	7.9	8	6.4	**0.69**
No	175	58	92.1	117	93.6	
**Fever**						
Yes	48	26	41.3	22	17.6	0.01 *****
No	140	37	58.7	103	82.4	
**Weight loss**						
Yes	33	14	22.2	19	15.2	0.23
No	155	49	77.8	106	84.8	
**Respiratory distress**						
Yes	16	5	7.9	11	8.8	0.84
No	172	58	92.1	114	91.2	
**Decrease in food consumption**						
Yes	56	24	38.1	32	25.6	0.09
No	132	39	61.9	93	74.4	
**Increase in the size of retropharyngeal nodes**						
Yes	57	33	52.4	24	19.2	0.00 *
No	131	30	47.6	101	80.8	
**Eye discharge**						
Yes	10	2	3.2	8	6.4	0.35
No	178	61	96.8	117	93.6	
**Decreased performance**						
Yes	51	22	34.9	29	23.2	0.09
No	137	41	65.1	96	76.8	

* statistically significant *p* value.

**Table 4 viruses-16-00839-t004:** Herd variables and health management.

Disinfection	%
Sodium hypochlorite	27.1% (51/188)
Creolin	22.3% (42/188)
Do not disinfect	13.8% (26/188)
Detergent	12.8% (24/188)
Ammonium 10%	11.7% (22/188)
Glutaraldehyde	9.0% (17/188)
Lime	3.2% (6/188)
Do not disinfect	13.8% (26/188)
Detergent	12.8% (24/188)
**Veterinary management**	**%**
On request if a problem occur	54.8% (103/188)
Permanent	45.2% (85/188)

**Table 5 viruses-16-00839-t005:** Bivariate and multivariate analysis of equine influenza virus symptoms and risk factors.

Variables	CRUDE OR	95% CI		*p*-Value	OR^adjusted using GEE^	95% CI		*p*-Value
Do not isolate sick from healthy animals	13.66	1.52	122.2	0.19	8.16	1.52	43.67	0.014 *
Share space with poultry	2.43	1.26	4.68	0.007	2.16	1.09	4.26	0.027 *
Live on property with equines that participate in equestrian events	2.57	1.25	5.28	0.01	1.32	0.612	2.86	0.47
Competition equine	2.7	1.22	5.99	0.12	1.92	0.75	4.9	0.17
Do not quarantine	1.90	1.01	3.54	0.044	1.88	0.41	8.52	0.16

* statistically significant *p* value.

## Data Availability

All the data are presented in the paper.

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
