# Peer review of "First Molecular Detection and Epidemiological Analysis of Equine Influenza Virus in Two Regions of Colombia, 2020–2023"

_viruses, 2024, doi:10.3390/v16060839_

Round 1

Reviewer 1 Report

Comments and Suggestions for Authors

Gonzalez-Obando et al. describe the detection, epidemiology, and characterization of EIV in horses in Colombia. The manuscript is interesting, and the authors identified avoidable infection risk factors, pointing out how infection rates could be effectively reduced. However, I have some doubts about the statistical analyses performed and I have some suggestions about data presentation that could improve the quality of the manuscript. The language is overall ok, but there are a few typos and errors that need to be corrected. Specific comments:

1. I am struggling to understand the relevance that “living in contact with other animals” has and I think that this aspect should be clarified. First, in the introduction, you should include: how equid H3N8 viruses originated from bird viruses, cross-species transmission and generation of canine H3N8 viruses, occurrence and frequency of cross-species transmission between horses and other animals. You also have to justify why you include contacts with other animals in your regression analysis and give it so much importance in the text. Was there any reason to expect zoonoses?

2. Results are a bit confusing, and the section could be improved by a better organization with fewer paragraphs, more tables, and a more streamlined analysis separating better herd characteristics, prevention practices at farms, infection risk factors, and infection outcomes.

3. Were age and the number of animals in herds normally distributed? If not, you should use the median and IQR instead of the average and SD.  

4. Sequences need to be submitted to Genbank and accession numbers provided.

5. At line 141, did you mean P<0.05? Also, confidence levels for what?

6. Additional details about the statistical analyses should be included. For example, the bivariate analyses and ORs are not mentioned in the M&M and for the multivariable model, it is not clear which factors were included. Did you just include all possible variables, or did you select those that showed an association (e.g., in a step-by-step way)? Authors should keep in mind that overcomplicating the model may result in reduced confidence and false positive significant associations… More importantly, it is not clear to me why you combined infection risk factors and clinical signs in the same model. What was the goal of this analysis? If you are trying to assess what are the risk factors for an infection, it is unclear to me why you’d consider enlarged lymph nodes important. I don’t think it is correct to keep clinical outcomes together with epidemiological risk factors when assessing infection risk. I strongly suggest re-evaluating this approach and building a more reasoned model.

7. Please, elaborate a little bit more on the sequencing part. How were the samples for sequencing selected? Was it a single-tube 8-segment PCR or was just the HA amplified? Was it full segment amplification? Were PCRs performed with barcoded primers or were barcodes added at a later stage? Which polymerase was used for amplification?

8. In the results, more details need to be included about the sequencing. Were the 3 sequences from both regions? From how many herds? Can you say that it was the same strain causing all infections? Why only 3 sequences? Are they representative of the whole outbreak? Was there an epidemiological link between infected animals/herds?

9. Paragraphs 3.2-3.6. Were these differences significant? Tables about the prevalence in the various groups (males, females, competition activity, vaccination….) should be provided and a proper statistical analysis (Chi-square) performed. Maybe a separate table for clinical characteristics (symptoms and treatment) can also be provided. Since the positives were all from 2022, presumably from a subset of the herds, you could also do a parallel analysis for the affected herds alone. Were any herds/animals re-tested throughout the years? If yes, how was this considered in the analyses?

Minor:

- Line 42. “Orthomyxoviridae” should be written in italics.

- Line 43. The genus name is “Alphainfluenzavirus”, and the species name is “Alphainfluenzavirus influenzae”. Please, clarify.

- Line 81. Please, refer to this as “supplementary table 1”.

- Lines 124-131. Please, revise the grammar in this sentence.

- Table 1. Could you also add the number of positive herds?

- Lines 210-221. There seem to be a lot of repetitions between these 2 paragraphs.

- Lines 219-221. I do not understand what you mean here…

- Lines 217-21. Instead of ORs (which is addressed in paragraph 3.11), I would do a chi analysis here to see if this particular feature is associated with EIV positivity, and then pick up the OR in the 3.11 paragraph.

- Supplementary table 2, first line: is 0.12 correct?

- Lines 325-332. Please, revise the grammar of this sentence. Additionally, citations are needed.

- Lines 369-371. Why is this important if you think that the virus came from other horses?

- Lines 374-9. Please, revise the grammar in this sentence. 

Comments on the Quality of English Language

See specific comments to authors. 

Author Response

Dear Reviewer #1.

Please find below our point by point responses to the comments regarding our Manuscript. The changes are highlighted in Yellow in the file.

We would like to thank the Reviewers for their helpful suggestions, for critical analysis of the manuscript, and for providing new discussion topics.

Gonzalez-Obando et al. describe the detection, epidemiology, and characterization of EIV in horses in Colombia. The manuscript is interesting, and the authors identified avoidable infection risk factors, pointing out how infection rates could be effectively reduced. However, I have some doubts about the statistical analyses performed and I have some suggestions about data presentation that could improve the quality of the manuscript. The language is overall ok, but there are a few typos and errors that need to be corrected. Specific comments:

  1. I am struggling to understand the relevance that “living in contact with other animals” has and I think that this aspect should be clarified. First, in the introduction, you should include: how equid H3N8 viruses originated from bird viruses, cross-species transmission and generation of canine H3N8 viruses, occurrence and frequency of cross-species transmission between horses and other animals. You also have to justify why you include contacts with other animals in your regression analysis and give it so much importance in the text. Was there any reason to expect zoonoses?

R/.  Information was first included on EIV for the equine population. Line (48-49) was also mentioned in the introduction of studies in which equine influenza infection in canines and felines has been reported. (52-59). Also in the methodology and in the discussion  the importance and need to include the variables in order to evaluate their relationship was mentioned (lines 105-106)

  1. Results are a bit confusing, and the section could be improved by a better organization with fewer paragraphs, more tables, and a more streamlined analysis separating better herd characteristics, prevention practices at farms, infection risk factors, and infection outcomes.

R/. The change was implemented by reducing the size of the paragraphs containing the information in the tables. In addition, a table was included to summarize information about disinfectant use on the premises and veterinary medical care. (Lines 280–284). Figure 2 describes the characteristics of the herds.

  1. Were age and the number of animals in herds normally distributed? If not, you should use the median and IQR instead of the average and SD.  

R/. The data do not have a normal distribution, the median is 3 years and the interquartile range is 23 (line 199)

  1. Sequences need to be submitted to Genbank and accession numbers provided.

R/. the sequences has been upload to the Genbank and the  accession number(s) for the nucleotide sequences are:  PP737049 - PP737050 - PP737051

  1. At line 141, did you mean P<0.05? Also, confidence levels for what?

R/. Both (P value and confidence interval) were analyzed in order to know which variables entered the model. (line 162)

  1. Additional details about the statistical analyses should be included. For example, the bivariate analyses and ORs are not mentioned in the M&M and for the multivariable model, it is not clear which factors were included. Did you just include all possible variables, or did you select those that showed an association (e.g., in a step-by-step way)? Authors should keep in mind that overcomplicating the model may result in reduced confidence and false positive significant associations… More importantly, it is not clear to me why you combined infection risk factors and clinical signs in the same model. What was the goal of this analysis? If you are trying to assess what are the risk factors for an infection, it is unclear to me why you’d consider enlarged lymph nodes important. I don’t think it is correct to keep clinical outcomes together with epidemiological risk factors when assessing infection risk. I strongly suggest re-evaluating this approach and building a more reasoned model.

R/. The model that was carried out was exploratory, including all the variables, not only the population but also the individual. Since the GEE model allows this type of adjustment of herd and clinical variables to be made, it provides a better work.lines 162-166

  1. Please, elaborate a little bit more on the sequencing part. How were the samples for sequencing selected? Was it a single-tube 8-segment PCR or was just the HA amplified? Was it full segment amplification? Were PCRs performed with barcoded primers or were barcodes added at a later stage? Which polymerase was used for amplification?

R/. The samples that presented amplification bands in the electrophoresis gel were selected; the name of the PCR is Optiflu. The barcodes were added during the preparation of libraries with the SQK-LSk109 and EXP-NBD196 kits from Oxford Nanopore Technologies and the polymerase used was, SuperScript™ III One-Step RT-PCR System with Platinum™ Taq DNA Polymerase (Ref. 12574026) lines (138-147)

  1. In the results, more details need to be included about the sequencing. Were the 3 sequences from both regions? From how many herds? Can you say that it was the same strain causing all infections? Why only 3 sequences? Are they representative of the whole outbreak? Was there an epidemiological link between infected animals/herds?

R/.  The sequences are from Antioquia, the region with the highest number of horses, which came from 3 different farms, these three sequences were the ones with the best quality in the amplification of the genome. The three properties belonged to one of the regions with the greatest equestrian activity in the Department (Antioquia) and the country.

  1. Paragraphs 3.2-3.6. Were these differences significant? Tables about the prevalence in the various groups (males, females, competition activity, vaccination….) should be provided and a proper statistical analysis (Chi-square) performed. Maybe a separate table for clinical characteristics (symptoms and treatment) can also be provided. Since the positives were all from 2022, presumably from a subset of the herds, you could also do a parallel analysis for the affected herds alone. Were any herds/animals re-tested throughout the years? If yes, how was this considered in the analyses?

R/. Table number 2 was complemented with variables, gender and type of activity. The other variables (epidemiological and clinical) and the chi-square value are in supplementary table 2. An analysis of the positive properties and their characteristics was carried out. In addition, the analysis was carried out by positive properties, relating it to the frequency of the analyzed categories. which was included in lines (297-316). Also in table number 1  the positive properties were added.

Minor:

- Line 42 “Orthomyxoviridae” should be written in italics. The suggestion was accepted 

- Line 43. The genus name is “Alphainfluenzavirus”, and the species name is “Alphainfluenzavirus influenzae”. Please, clarify. The suggestion was accepted 

- Line 81. Please, refer to this as “supplementary table 1”. The suggestion was accepted,

- Lines 124-131. Please, revise the grammar in this sentence.  The suggestion was accepted 

- Table 1. Could you also add the number of positive herds? The suggestion was accepted table 1

- Lines 210-221. There seem to be a lot of repetitions between these 2 paragraphs.:   The suggestion was accepted, repeated ideas were eliminated. 

- Lines 219-221. I do not understand what you mean here…. the meaning of the phrase was organized. The suggestion was accepted. 

- Lines 217-21. Instead of ORs (which is addressed in paragraph 3.11), I would do a chi analysis here to see if this particular feature is associated with EIV positivity, and then pick up the OR in the 3.11 paragraph.

- Supplementary table 2, first line: is 0.12 correct?The suggestion was accepted

- Lines 325-332. Please, revise the grammar of this sentence. Additionally, citations are needed The suggestion was accepted, a reference was included. 

- Lines 369-371. Why is this important if you think that the virus came from other horses? This part was corrected.

- Lines 374-9. Please, revise the grammar in this sentence.  the meaning and the grammar of the phrase were organized. This part was corrected. 

Reviewer 2 Report

Comments and Suggestions for Authors

I completely agree with the authors that such research is necessary in a country where horse breeding is so developed and equine influenza causes significant damage. This article is important for the development of horse breeding. However, a major revision of the manuscript is needed.

Point 1. Keywords: Horses and equine influenza must be added as keywords.

Point 2. Figure 1 (map of Colombia) is meaningless. The expert would recommend moving it to supplements or removing it completely.

Point 3. Materials and Methods. It is unclear to the reader not familiar with Colombia (or perhaps even those who are familiar with it) what kind of horses we are talking about - domestic horses on stud farms, farms, or wild horses? If these are domestic animals, you should indicate not only the number of livestock surveyed, but also the number of farms on which the research was carried out.

Point 4. Figure 3. It seems to me that this comparison is not entirely correct. Are you talking about animals susceptible to the influenza A virus? But of the four animal species presented in Figure 3, only two are actually sensitive to influenza A viruses (these are pigs and birds); dogs and cats are accidental hosts and are not capable of transmitting the virus person-to-person. All of the above, as well as the alpha2,3 and alpha2,6 specificity of the species presented, need to be discussed.

Or are the authors talking about sensitivity to the equine influenza virus? Where are the literary references that support this?

Also, what type of receptor specificity do horses have? Alpha2.3 or alpha2.6? The authors don't mention this anywhere. What did they want to find? Transmissibility cases of equine virus to cats from horses or from horses to cats?

If, when sick and healthy horses are kept together, the latter can become infected from the former, this is absolutely understandable. What biological sense does it make to compare animals housed with dogs or cats (paragraph 3.5, Figure 3)? Why did this question arise? Are the authors suggesting that the equine virus is transmitted from cats? What is the point of the study? The authors call this “Coexistence of horses with other animal species susceptible to influenza A viruses,” but neither cats nor dogs are susceptible to influenza A viruses, and especially to equine influenza. Yes, the scientific literature describes isolated cases of infection of cats, both wild and domestic, and dogs with H5N1 avian influenza viruses, but what does this have to do with equine influenza? These animals are so-called accidental hosts. It is necessary to discuss the alpha2,3 and alpha2,6 specificity of the presented species. As mentioned above, dogs and cats are accidental hosts and are not capable of transmitting the influenza virus person-to-person.

The results obtained in Section 3.5 were not discussed in the Discussion section at al. Only in the very last section - Conclusion - the authors mentioned that coexistence with birds is bad, but in the Results there is no confirmation of this. The authors did not examine dogs, cats, pigs and birds for the presence of influenza virus. Birds (it is not even said which particular kind of birds - poultry or, for example, sparrows, crows, i.e. wild birds that live near horses and feed, for example, on horse food) have not been examined for carriage of avian influenza viruses, and the sensitivity of horses to avian influenza has not been documented by the authors’ studies or literary data. In addition, it is not indicated which specific birds are meant?

I believe that section 3.5 does not provide any scientific information and should be completely rewritten taking into account the comments made or deleted.

 Point 5. Line 212-213. I think “one the other hand” in mistake and should be replaced with “on the other hand.”

Author Response

Dear Reviewer #2.

Please find below our point by point responses to the comments regarding our Manuscript. The changes are highlighted in Yellow in the file.

We would like to thank the Reviewers for their helpful suggestions, for critical analysis of the manuscript, and for providing new discussion topics.

I completely agree with the authors that such research is necessary in a country where horse breeding is so developed and equine influenza causes significant damage. This article is important for the development of horse breeding. However, a major revision of the manuscript is needed.

Point 1. Keywords: Horses and equine influenza must be added as keywords.

R/. Horses and equine  were added to the keywords line (38)

Point 2. Figure 1 (map of Colombia) is meaningless. The expert would recommend moving it to supplements or removing it completely.

R/. The map was added to the supplementary material.

Point 3. Materials and Methods. It is unclear to the reader not familiar with Colombia (or perhaps even those who are familiar with it) what kind of horses we are talking about - domestic horses on stud farms, farms, or wild horses? If these are domestic animals, you should indicate not only the number of livestock surveyed, but also the number of farms on which the research was carried out.

R/. The type of horses included was clarified, which were domestic. Besides, the number of farms or herds was included in table number 1. the importance and need to include the variables in order to evaluate their relationship was mentioned (lines 105-106)

Point 4. Figure 3. It seems to me that this comparison is not entirely correct. Are you talking about animals susceptible to the influenza A virus? But of the four animal species presented in Figure 3, only two are actually sensitive to influenza A viruses (these are pigs and birds); dogs and cats are accidental hosts and are not capable of transmitting the virus person-to-person. All of the above, as well as the alpha2,3 and alpha2,6 specificity of the species presented, need to be discussed

R/. First, the data was added to the horses' alpha 2 and 3 receptors in line (48). Furthermore, the influenza virus type A's affinity for birds, pigs, dogs, and cats was explained lines 52-59

Or are the authors talking about sensitivity to the equine influenza virus? Where are the literary references that support this?

R/. The sensitivity of the equine influenza virus to other species was included in lines 52-59.

Also, what type of receptor specificity do horses have? Alpha2.3 or alpha2.6? The authors don't mention this anywhere. What did they want to find? Transmissibility cases of equine virus to cats from horses or from horses to cats?

R/. Information on the type of alpha 3 receptor in horses was included in line 48. Our objective was to be able to see the relationship between the presence of equine influenza and other animal populations (canines, felines, and birds). in lines 52-59.

If, when sick and healthy horses are kept together, the latter can become infected from the former, this is absolutely understandable. What biological sense does it make to compare animals housed with dogs or cats (paragraph 3.5, Figure 3)? Why did this question arise? Are the authors suggesting that the equine virus is transmitted from cats? What is the point of the study? The authors call this “Coexistence of horses with other animal species susceptible to influenza A viruses,” but neither cats nor dogs are susceptible to influenza A viruses, and especially to equine influenza. Yes, the scientific literature describes isolated cases of infection of cats, both wild and domestic, and dogs with H5N1 avian influenza viruses, but what does this have to do with equine influenza? These animals are so-called accidental hosts. It is necessary to discuss the alpha2,3 and alpha2,6 specificity of the presented species. As mentioned above, dogs and cats are accidental hosts and are not capable of transmitting the influenza virus person-to-person.

R/. It has been demonstrated in literature, which we refer to, that equine influenza infection is in canines and felines, which share alpha 3 and alpha 6 receptors. That is why other research also includes the analysis of these species in order to recognize other hosts as a source of interspecies transmission or risk. Our intention was to analyze it among animals, and not in human subjects. Therefore the clarification was made in introduction, methodology and discussion. in lines 52-59.

The results obtained in Section 3.5 were not discussed in the Discussion section at al. Only in the very last section - Conclusion - the authors mentioned that coexistence with birds is bad, but in the Results there is no confirmation of this. The authors did not examine dogs, cats, pigs and birds for the presence of influenza virus. Birds (it is not even said which particular kind of birds - poultry or, for example, sparrows, crows, i.e. wild birds that live near horses and feed, for example, on horse food) have not been examined for carriage of avian influenza viruses, and the sensitivity of horses to avian influenza has not been documented by the authors’ studies or literary data. In addition, it is not indicated which specific birds are meant?

R/. We included what type of birds (Poultry) lines (35,248,249,251,253,254)and this adjustment was made in the text. It was also included in the discussion lines (411-415)

I believe that section 3.5 does not provide any scientific information and should be completely rewritten taking into account the comments made or deleted.

R/. We included scientific support for the species barrier jump in the introduction, materials, and methods, as well as the affinity of the species' receptors (birds, canines, felines,). We also adjusted session 3.5 to be related to the introduction in lines 52-59., materials and methods lines 105-106

 Point 5. Line 212-213. I think “one the other hand” in mistake and should be replaced with “on the other hand.”

R/.  The suggestion was accepted

Round 2

Reviewer 1 Report

Comments and Suggestions for Authors

I thank the authors for implementing the required revisions. The manuscript is now clearer and more complete. However, there are still a few major issues that need to be resolved before I can recommend manuscript publication.

1. the text needs some language revisions as there are a lot of errors and several unclear sentences.

2. I am a bit confused about the sequence analyses. Specifically:

- Why was the complete genome obtained and only the HA analyzed?

- If you sequenced 3/63 infected horses, which were only from one region, how can you tell they were all infected by the same virus? You should have sequence information at least for most herds to conclude that it’s the same virus/lineage to cause all infections.

- in the methods you say you built a maximum likelihood tree, but the tree caption says maximum credibility (Bayesian analysis).

3. I think the authors should revise the statistical analysis. Specifically:

- Line 158. Was 0.25 really the cut-off you used?

- Table 2. 8/27 is not 70%, but 29.6. Therefore, the age group with the highest prevalence was <2 years.

- Line 305. If I understood your study design correctly, you do not do a bivariate analysis, but a monovariate.

- You say “chi-square or Fisher's exact tests there is nothing in the text”, but I did not see any results of this analysis, which could nicely be used in Table 2 and to compare symptoms in infected vs non-infected horses.

- The fact that a software allows you to include all the variables does not mean that it makes sense to do so. The software will give you an OR for everything, even for variables that have no reason to be included. Specifically, I find bizarre the idea of considering clinical outcomes and medications as infection risks! In your model, you are assessing what increases the risk of becoming infected, and certain symptoms, which manifest after you are infected, are not risk factors! If you want to find out which are the symptoms associated to EIV, you do a Chi test comparing infected and not infected, but you can’t include them in your model that wants to look at infection risks (those that favor the infection). In your model of risk assessment, you should only include those variables that make sense to be included, like age, travel, vaccination status, quarantine, etcetera. Including clinical outcomes and therapies makes no sense. Even for the potential risk factors, these should be selected based on actual risk hypotheses. Why do you think it is important to include contact with chickens? Is there any evidence that EIV transmits from horse to chicken and then back to horses? If not, you are simply assuming that horses are getting AIV from chickens, but you sequenced the virus, and this is not the case. So, what is the hypothesis here? I recommend the authors build a more reasoned model, including only variables hypothesized to be associated with an actual infection increase risk. Sentences like ”The odds of testing EIV positive was 14 times higher in horses housed in herds where sick animals were not separated from healthy individuals after adjusting for the effect of fever, enlarged lymph nodes and shared space with poultry” make no sense. How does an enlarged lymph node affect the risk of acquiring EIV if it’s a symptom of the already-acquired infection?

- Lines 235-8. If you conclude that horses were infected by equine influenza this analysis makes no sense because the animals would have acquired the infection from other horses (which is what I think your message is). If this is the case, this association is likely spurious (maybe linked to farming practices?).

Minor:

- Lines 197-200. This part is not clear.

- Line 205. Males are in there twice

Comments on the Quality of English Language

the text needs some language revisions as there are a lot of errors and several unclear sentences.

Author Response

Thank the authors for implementing the required revisions. The manuscript is now clearer and more complete. However, there are still a few major issues that need to be resolved before I can recommend manuscript publication.

  1. the text needs some language revisions as there are a lot of errors and several unclear sentences.

R/. The full text was properly edited by using Curie™ Part of SpringerNature™

  1. I am a bit confused about the sequence analyses. Specifically:

- Why was the complete genome obtained and only the HA analyzed?

R/. We agree to the reviewer. However, The Aim of the sequencing was only to confirm the presence of H3 viruses on the Country. Not to develop evolutive analysis. The complete genome and evolutive analysis requires a full new Paper that are in development including more sequences.

- If you sequenced 3/63 infected horses, which were only from one region, how can you tell they were all infected by the same virus? You should have sequence information at least for most herds to conclude that it’s the same virus/lineage to cause all infections.

R/. We partially agree with the reviewer. Currently, only one serotype (H3N8) is circulating globally since the H7N7 virus is considered extinct. Furthermore, it is well-known that the evolution of EIV is significantly slower than that of other mammalian influenza viruses (https://doi.org/10.1128/JVI.02619-10). The detection of H3 viruses in the Antioquia Department and the positive qPCR results in the samples confirm the presence of Equine Influenza Virus. However, we have added a brief disclaimer in the Discussion section regarding the need for extensive phylogenetic studies to explain the molecular evolution of the virus in Colombia.

- in the methods you say you built a maximum likelihood tree, but the tree caption says maximum credibility (Bayesian analysis).

R/. We agree to the reviewer and apologize for this unintentional mistake.

  1. I think the authors should revise the statistical analysis. Specifically:

- Line 158. Was 0.25 really the cut-off you used?  

R/ The cut-off points to evaluate association was 0.05. and finally, those variables with a p value less than 0.25 entered the model. (Lines 166-168)

- Table 2. 8/27 is not 70%, but 29.6. Therefore, the age group with the highest prevalence was <2 years.

R/ Sorry for the mistake, we already made the change in table 2. We also adjust on line 220, on the age group with the highest number of infected individuals, those under 2 years old.

- Line 305. If I understood your study design correctly, you do not do a bivariate analysis, but a monovariate.

R/In our study we do a monovariate, bivariate and multivariate analysis. We apologize for not including the bivariate information in the body of the article. To do this, we include the bivariate information of the clinical variables in table number 3 (Line 227). The bivariate of the epidemiological or risk factor variables are part of the first 3 columns of table number 5. (Line 353)

- You say “chi-square or Fisher's exact tests there is nothing in the text”, but I did not see any results of this analysis, which could nicely be used in Table 2 and to compare symptoms in infected vs non-infected horses.

R/ We include the bivariate information of the clinical variables, with the P values, corresponding to the result of the chi-square and fisher test in table number 3 (Line 237) and in lines 227-235. which is similar to table two, but is only about symptoms and statistical association, with the positivity of the molecular diagnosis, The information corresponding to the other variables and the association, is in supplementary table 2.

- The fact that a software allows you to include all the variables does not mean that it makes sense to do so. The software will give you an OR for everything, even for variables that have no reason to be included. Specifically, I find bizarre the idea of considering clinical outcomes and medications as infection risks! In your model, you are assessing what increases the risk of becoming infected, and certain symptoms, which manifest after you are infected, are not risk factors! If you want to find out which are the symptoms associated to EIV, you do a Chi test comparing infected and not infected, but you can’t include them in your model that wants to look at infection risks (those that favor the infection). In your model of risk assessment, you should only include those variables that make sense to be included, like age, travel, vaccination status, quarantine, etcetera. Including clinical outcomes and therapies makes no sense. Even for the potential risk factors, these should be selected based on actual risk hypotheses. Why do you think it is important to include contact with chickens? Is there any evidence that EIV transmits from horse to chicken and then back to horses? If not, you are simply assuming that horses are getting AIV from chickens, but you sequenced the virus, and this is not the case. So, what is the hypothesis here? I recommend the authors build a more reasoned model, including only variables hypothesized to be associated with an actual infection increase risk. Sentences like” The odds of testing EIV positive was 14 times higher in horses housed in herds where sick animals were not separated from healthy individuals after adjusting for the effect of fever, enlarged lymph nodes and shared space with poultry” make no sense. How does an enlarged lymph node affect the risk of acquiring EIV if it’s a symptom of the already-acquired infection?

R/. We agree with the comment and apologize for the earlier misunderstanding. We appreciate your persistence. We now clearly see that the clinical variables preceding an equine influenza infection have been excluded. We have retained the epidemiological factors in the model that support the virus's existence. Therefore, the final association model was built solely with the exposure variables. Living with birds might indicate poor bio-sanitary management on farms, as mentioned in the text (lines 441-445). We included this variable because equine influenza is of avian origin (line 54).

- Lines 235-8. If you conclude that horses were infected by equine influenza this analysis makes no sense because the animals would have acquired the infection from other horses (which is what I think your message is). If this is the case, this association is likely spurious (maybe linked to farming practices?).

R/We agree with the reviewer. The possibility of greater infection in those horses that lived with birds may be related to poor biosecurity practices. which was clarified in lines (441-445)

Minor:

- Lines 197-200. This part is not clear.

R/ We corrected the error, in writing. between line (212-216)

- Line 205. Males are in there twice:

R/ sorry for repeated information, it was corrected and synthesized on the lines (212-216)

Reviewer 2 Report

Comments and Suggestions for Authors

The manuscript has been properly revised according to the Reviewer's comments. The authors answered all the comments of the Reviewer.

Author Response

Thanks for your helpful review.